# Rethinking the Structure of Stochastic Gradients: Empirical and Statistical Evidence

## Abstract

It is well known that stochastic gradients significantly improve both optimization and generalization of deep neural networks (DNNs). Some works attempted to explain the success of stochastic optimization for deep learning by the arguably heavy-tail properties of gradient noise, while other works presented theoretical and empirical evidence against the heavy-tail hypothesis on gradient noise. Unfortunately, formal statistical tests for analyzing the structure and heavy tails of stochastic gradients in deep learning are still under-explored. In this paper, we mainly make two contributions. First, we conduct formal statistical tests on the distribution of stochastic gradients and gradient noise across both parameters and iterations. Our statistical tests reveal that dimension-wise gradients usually exhibit power-law heavy tails, while iteration-wise gradients and stochastic gradient noise caused by minibatch training usually do not exhibit power-law heavy tails. Second, we further discover that the covariance spectra of stochastic gradients have the power-law structures in deep learning. While previous papers believed that the anisotropic structure of stochastic gradients matters to deep learning, they did not expect the gradient covariance can have such an elegant mathematical structure. Our work challenges the existing belief and provides novel insights on the structure of stochastic gradients. The novel structure of stochastic gradients may help understand the success of stochastic optimization for deep learning.

## 1 Introduction

Stochastic optimization methods, such as Stochastic Gradient Descent (SGD), have been highly successful and even necessary in the training of deep neural networks (LeCun et al., 2015). It is widely believed that stochastic gradients as well as stochastic gradient noise (SGN) significantly improve both optimization and generalization of deep neural networks (DNNs) (Hochreiter & Schmidhuber, 1995; 1997; Hardt et al., 2016; Wu et al., 2021; Smith et al., 2020; Wu et al., 2020; Sekhari et al., 2021; Amir et al., 2021). SGN, defined as the difference between full-batch gradient and stochastic gradient, has attracted much attention in recent years. People studied its type (Simsekli et al., 2019; Panigrahi et al., 2019; Hodgkinson & Mahoney, 2021; Li et al., 2021), its magnitude (Mandt et al., 2017; Liu et al., 2021), its structure (Daneshmand et al., 2018; Zhu et al., 2019; Chmiel et al., 2020; Xie et al., 2020; Wen et al., 2020), and its manipulation (Xie et al., 2021). Among them, the noise type and the noise covariance structure are two core research topics.

**Topic 1. The arguments on the type and the heavy-tailed property of SGN.** Recently, a line of research (Simsekli et al., 2019; Panigrahi et al., 2019; Gurbuzbalaban et al., 2021; Hodgkinson & Mahoney, 2021) argued that SGN has the heavy-tail property due to Generalized Central Limit Theorem (Gnedenko et al., 1954). Simsekli et al. (2019) presented statistical evidence showing that SGN looks closer to an $\alpha$-stable distribution that has *power-law heavy tails* rather than a Gaussian distribution. (Panigrahi et al., 2019) also presented the Gaussianity tests. However, their statistical tests were actually not applied to the true SGN that is caused by minibatch sampling. Because, in this line of research, the abused notation "SGN" is studied as stochastic gradient at some iteration rather than the difference between full-batch gradient and stochastic gradient. Another line of research (Xie et al., 2020; 2022b; Li et al., 2021) pointed out this issue and suggested that the arguments in Simsekli et al. (2019) rely on a hidden strict assumption that SGN must be isotropic and does not hold for *parameter-dependent and anisotropic* Gaussian noise. This is why *one tail-index for all parameters*

was studied in Simsekli et al. (2019). In contrast, SGN could be well approximated as an *multi-variant* Gaussian distribution in experiments at least when batch size is not too small, such as $B \geq 128$ (Xie et al., 2020; Panigrahi et al., 2019). Another work (Li et al., 2021) further provided theoretical evidence for supporting the anisotropic Gaussian approximation of SGN. Nevertheless, none of these works conducted statistical tests on the Gaussianity or heavy tails of the true SGN.

**Contribution 1.** To our knowledge, we are the first to conduct formal statistical tests on the distribution of stochastic gradients/SGN across parameters and iterations. Our statistical tests reveal that dimension-wise gradients (due to anisotropy) exhibit power-law heavy tails, while iteration-wise gradient noise (which is the true SGN due to minibatch sampling) often has Gaussian-like light tails. Our statistical tests and notations help reconcile recent conflicting arguments on Topic 1.

**Topic 2. The covariance structure of stochastic gradients/SGN.** A number of works (Zhu et al., 2019; Xie et al., 2020; HaoChen et al., 2021; Liu et al., 2021; Ziyin et al., 2022) demonstrated that the anisotropic structure and sharpness-dependent magnitude of SGN can help escape sharp minima efficiently. Moreover, some works theoretically demonstrated (Jastrzkebski et al., 2017; Zhu et al., 2019) and empirically verified (Xie et al., 2020; 2022b; Daneshmand et al., 2018) that the covariance of SGN is approximately equivalent to the Hessian near minima. However, this approximation is only applied to minima and along flat directions corresponding to nearly-zero Hessian eigenvalues. The covariance structure of stochastic gradients is still a fundamental issue in deep learning.

**Contribution 2.** We discover that the covariance of stochastic gradients has the power-law spectra in deep learning, while full-batch gradients have no such properties. While previous papers believed that the anisotropic structure of stochastic gradients matters to deep learning, they did not expect the gradient covariance can have such an elegant power-law structure. The power-law covariance may help understand the success of stochastic optimization for deep learning.

**Organization.** In Section 2, we introduce prerequisites and the statistical test methods. In Section 3, we reconcile the conflicting arguments on the noise type and heavy tails in SGD. In Section 4, we discover the power-law covariance structure. In Section 5, we present extensive empirical results for deeply exploring the covariance structure. In Section 6, we conclude our main contributions.

## 2 METHODOLOGY: NOTATIONS AND GOODNESS-OF-FIT TESTS

**Notations.** Suppose a neural network $f_\theta$ has $n$ model parameters as $\theta$. We denote the training dataset as $\{(x, y)\} = \{(x_j, y_j)\}_{j=1}^N$ drawn from the data distribution $\mathcal{S}$ and the loss function over one data sample $\{(x_j, y_j)\}$ as $l(\theta, (x_j, y_j))$. We denote the training loss as $L(\theta) = \frac{1}{N} \sum_{j=1}^N l(\theta, (x_j, y_j))$.

We compute the gradients of the training loss with the batch size $B$ and the learning rate $\eta$ for $T$ iterations. We let $g^{(t)}$ represent the stochastic gradient at the $t$-th iteration. We denote the *Gradient History Matrix* as $G = [g^{(1)}, g^{(2)}, \cdots, g^{(T)}]$ an $n \times T$ matrix where the column vector $G_{\cdot,t}$ represents the *dimension-wise gradients* $g^{(t)}$ for $n$ model parameters, the row vector $G_{i,\cdot}$ represents the *iteration-wise gradients* $g_i$ for $T$ iterations, and the element $G_{i,t}$ is $g_i^{(t)}$ at the $t$-th iteration for the parameter $\theta_i$. We analyze $G$ for a given model without updating the model parameter $\theta$. The Gradient History Matrix $G$ plays a key role in reconciling the conflicting arguments on Topic 1. Because the defined dimension-wise SGN (due to anisotropy) is the abused "SGN" in one line of research (Simsekli et al., 2019; Panigrahi et al., 2019; Gurbuzbalaban et al., 2021; Hodgkinson & Mahoney, 2021), while iteration-wise SGN (due to minibatch sampling) is the true SGN as another line of research (Xie et al., 2020; 2022b; Li et al., 2021) suggested. Our notation mitigates the abused "SGN".

We further denote the second moment as $C_m = \mathbb{E}[gg^\top]$ for stochastic gradients and the covariance as $C = \mathbb{E}[(g - \bar{g})(g - \bar{g})^\top]$ for SGN, where $\bar{g} = \mathbb{E}[g]$ is the full-batch gradient. We denote the descending ordered eigenvalues of a matrix, such as the Hessian $H$ and the covariance $C$, as $\{\lambda_1, \lambda_2, \ldots, \lambda_n\}$ and denote the corresponding spectral density function as $p(\lambda)$.

**Goodness-of-Fit Test.** In statistics, various Goodness-of-Fit Tests have been proposed for measuring the goodness of empirical data fitting to some distribution. In this subsection, we introduce how to conduct the Kolmogorov-Smirnov (KS) Test (Massey Jr, 1951; Goldstein et al., 2004) for measuring the goodness of fitting a power-law distribution and the Pearson's $\chi^2$ Test (Plackett, 1983) for measuring the goodness of fitting a Gaussian distribution. We present more details in Appendix B.

When we say a set of random variables (the elements or eigenvalues) is approximately power-law/Gaussian in this paper, we mean the tested set of data points can pass KS Tests for power-law distributions or $\chi^2$ Test for Gaussian distributions at the Significance Level 0.05. We note that, in all statistical tests of this paper, we set the Significance Level as 0.05.

In KS Test, we state *the power-law hypothesis* that the tested set of elements is power-law. If the KS distance $d_{ks}$ is larger than the critical distance $d_c$, the KS test will reject the power-law hypothesis. In contrast, if the KS distance $d_{ks}$ is less than the critical distance $d_c$, the KS test will support (not reject) the power-law hypothesis. The smaller $d_{ks}$ is, the better the goodness-of-power-law is.

In $\chi^2$ Test, we state *the Gaussian hypothesis* that the tested set of elements is Gaussian. If the estimated $p$-value is larger than 0.05, the $\chi^2$ test will reject the Gaussian hypothesis. If the estimated $p$-value is less than 0.05, the $\chi^2$ test will support (not reject) the Gaussian hypothesis. The smaller $p$-value is, the better the goodness-of-Gaussianity is.

The Gaussianity test consists of Skewness Test and Kurtosis Test (Cain et al., 2017) (See Appendix B). Skewness is a measure of symmetry. A distribution or data set is symmetric if the distribution on either side of the mean value is roughly the mirror image of the other. Kurtosis is a measure of whether the data are heavy-tailed or light-tailed relative to a normal distribution. Empirical data with high kurtosis tend to have heavy tails. Empirical data with low kurtosis tend to have light tails. Thus, $\chi^2$ Test can reflect both Gaussianity and heavy tails.

## 3 RETHINK HEAVY TAILS IN STOCHASTIC GRADIENTS

In this section, we try to reconcile the conflicting arguments on Topic 1 by formal statistical tests.

**The power-law distribution.** Suppose that we have the set of $k$ random variables $\{\lambda_1, \lambda_2, \ldots, \lambda_k\}$ that obeys a power-law distribution. We may write the probability density function of a power-law distribution as

$$p(\lambda) = Z^{-1}\lambda^{-\beta}, \tag{1}$$

where $Z$ is the normalization factor. The finite-sample power law, also known as Zipf's law, can also be approximately written as

$$\lambda_k = \lambda_1 k^{-s}, \tag{2}$$

if we let $s = \frac{1}{\beta-1}$ denote the power exponent of Zipf's law (Visser, 2013). A well-known property of power laws is that, when the power-law variables and their corresponding rank orders are scattered in log-log scaled plot, a straight line may fit the points well (Clauset et al., 2009).

**Dimension-wise gradients are usually power-law, while iteration-wise gradients are usually Gaussian.** In Figure 1, we plot dimension-wise gradients and iteration-wise gradients of LeNet on MNIST, CIFAR-10, and CIFAR-100 over 5000 iterations with fixing the model parameters. We leave experimental details in Appendix A and the extensive statistical test results in Appendix C. In Figure 1, while some points slightly deviate from the fitted straight line, we may easily observe the straight lines approximately fit the red points (dimension-wise gradients) but fail to fit the blue points (iteration-wise gradients). The observations indicate that dimension-wise gradients have power-law heavy tails while iteration-wise gradients have no power-law heavy tails.

Table 1 shows the mean KS distance and the mean $p$-value over dimensions and iterations as well as the power-law rates and the Gaussian rates. We note that the power-law/Gaussian rate means the percentage of the tested points that are not rejected for the power-law/Gaussian hypothesis via KS/$\chi^2$ tests. Dimension-wise gradients and iteration-wise gradients show significantly different preferences for the power-law rate and the Gaussian rate. For example, a LeNet on CIFAR-10 has 62006 model parameters. Dimension-wise gradients of the model are power-law for $81.8\%$ iterations and are Gaussian for only $0.3\%$ iterations. In contrast, iteration-wise gradients of the model are Gaussian for $66.8\%$ dimensions (parameters) and are power-law for no dimension.

The observation and the statistical test results of Table 1 both indicate that dimension-wise gradients usually have power-law heavy tails while iteration-wise gradients are usually approximately Gaussian (with light tails) for most dimensions. The conclusion holds for both pretrained models and random models on various datasets. Similarly, we also observe power-law dimension-wise gradients and non-power-law iteration-wise gradients for FCN and ResNet18 in Figure 3 and Table 4 of Appendix C.

Table 1: The KS and $\chi^2$ statistics and the hypothesis acceptance rates of the gradients over dimensions and iterations, respectively. Model: LeNet. Batch Size: 100. In the second column "random" means randomly initialized models, while "pretrain" means pretrained models.

| Dataset | Training | SG Type | $\bar{d}_{\mathrm{ks}}$ | $d_c$ | Power-Law Rate | $\bar{p}$-value | Gaussian Rate |
|---------|----------|---------|--------|-------|----------------|---------|---------------|
| MNIST | Random | Dimension | 0.0355 | 0.0430 | 76.6% | $4.11 \times 10^{-4}$ | 0.18% |
| MNIST | Random | Iteration | 0.191 | 0.0430 | 0.0788% | 0.321 | 65.1% |
| MNIST | Pretrain | Dimension | 0.0401 | 0.0430 | 60.6% | $5.77 \times 10^{-4}$ | 0.24% |
| MNIST | Pretrain | Iteration | 0.179 | 0.0430 | 0.052% | 0.306 | 62.0% |
| CIFAR-10 | Random | Dimension | 0.0330 | 0.0430 | 81.8% | $1.03 \times 10^{-3}$ | 0.3% |
| CIFAR-10 | Random | Iteration | 0.574 | 0.0430 | 0% | 0.337 | 66.8% |
| CIFAR-10 | Pretrain | Dimension | 0.0381 | 0.0430 | 69.2% | $4.01 \times 10^{-5}$ | 0% |
| CIFAR-10 | Pretrain | Iteration | 0.654 | 0.0430 | 0% | 0.275 | 56.6% |

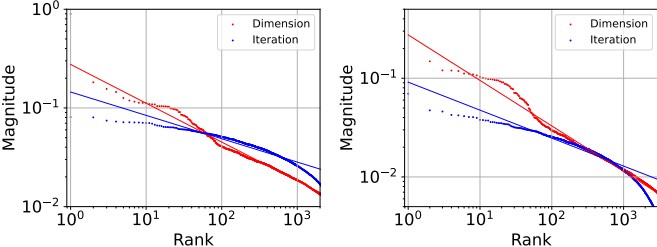 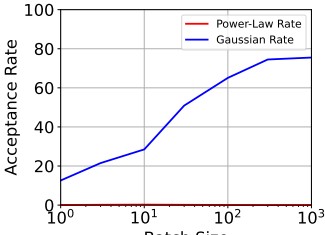

Figure 1: We plot the magnitude of gradients with respect to the magnitude rank. The dimension-wise gradients have power-law heavy tails, while the iteration-wise gradients have no power-law heavy tails. Model: Randomly Initialized LeNet. Dataset: MNIST and CIFAR-10.

Figure 2: The power-law rates and Gaussian rates w.r.t. the batch size. Increasing batch size significantly improves the Gaussianity of SGN. Model: LeNet.

According to Central Limit Theorem, the Gaussianity of iteration-wise gradients should depend on the batch size. We empirically studied how the Gaussian rate of iteration-wise gradients depends on the batch size. The results in Figure 2 and Table 3 support that the Gaussianity of iteration-wise gradients indeed positively correlates to the batch size, which is consistent with the Central Limit Theorem. In the common setting that $B \geq 30$, the Gaussianity of SGN can be statistically more significant than heavy tails for most parameters of DNNs, according to $\chi^2$ Tests.

**Reconciling the conflicting arguments on Topic 1.** We argue that the power-law tails of dimension-wise gradients and the Gaussianity of iteration-wise gradients may well explain the conflicting arguments on Topic 1. On the one hand, the evidences proposed by the first line of research are mainly for describing the elements of one column vector of $G$ which represent the dimension-wise gradient at a given iteration. Thus, the works in the first line of research can only support that the distribution of (dimension-wise) stochastic gradients has a power-law heavy tail, where heavy tails are mainly caused by the gradient covariance (See Section 4) instead of minibatch training.

On the other hand, the works in the second line of research pointed out that the type of SGN is actually decided by the distribution of (iteration-wise) stochastic gradients due to minibatch sampling, which is usually Gaussian for a common batch size $B \geq 30$. Researchers care more about the true SGN, the difference between full-batch gradients and stochastic gradients, mainly because SGN essentially matters to implicit regularization of SGD and deep learning dynamics. While previous works in the second line of research did not conduct statistical tests, our work fills the gap.

In summary, while it seems that the two lines of research have conflicting arguments on Topic 1, their evidences are actually not contradicted. We may easily reconcile the conflicts as long as the first line of research clarifies that the heavy-tail property describes dimension-wise gradients (not SGN), which corresponds to the column vector of $G$ instead of the row vector of $G$.

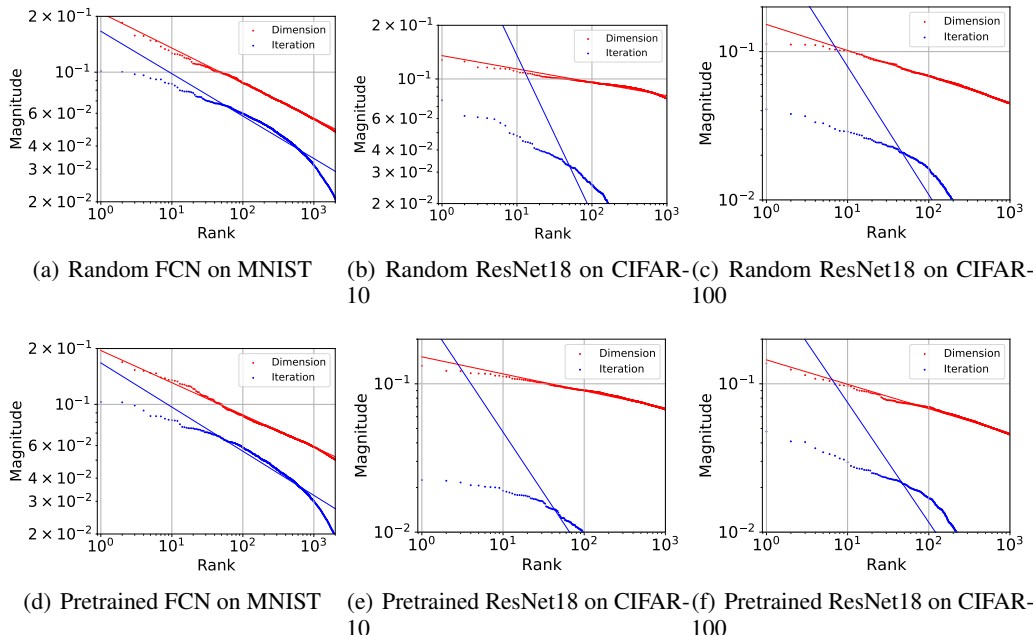

(a) Random FCN on MNIST   (b) Random ResNet18 on CIFAR-(c) Random ResNet18 on CIFAR-
10   100

(d) Pretrained FCN on MNIST   (e) Pretrained ResNet18 on CIFAR-(f) Pretrained ResNet18 on CIFAR-
10   100

Figure 3: We plot the magnitude of gradients with respect to the magnitude rank for FCN and ResNet18 on MNIST, CIFAR-10, and CIFAR-100. The dimension-wise gradients have power-law heavy tails, while the iteration-wise gradients have no power-law heavy tails.

We notice that the Gaussian rates (the rates of not rejecting the Gaussian hypothesis) do not approach to 100% even under relatively large batch sizes (e.g., $B = 1000$), while they have nearly zero power-law rates (the rates of not rejecting the power-law hypothesis). While the Gaussian rate is low under small batch sizes, the power-law rate is still nearly zero. This may indicate that SGN of a small number of model parameters or under small batch sizes may have novel properties beyond the Gaussianity and the power-law heavy tails that previous works expected.

## 4   THE POWER-LAW COVARIANCE OF STOCHASTIC GRADIENTS

In this section, we mainly studied the covariance/second-moment structure of stochastic gradients in deep learning. Despite the reconciled conflicts on Topic 1, another question arises that why dimension-wise stochastic gradients may exhibit power laws in deep learning. We show that the covariance not only explains why power-law gradients arise but also surprisingly challenges conventional knowledge on the relation between the covariance (of SGN) and the Hessian (of the training loss).

**The power-law covariance spectrum.** We first display the covariance spectra for various models on MNIST and CIFAR-10. Figure 4 shows that the covariance spectra for pretrained models and random models are both power-law despite several slightly deviated top eigenvalues. The KS test results are shown in Table 2. To our knowledge, we are the first to discover that the covariance spectra are usually power-law for various DNNs with formal empirical and statistical evidences.

**The relation between the covariance and the Hessian.** The relation between the covariance and the Hessian is interesting because both SGN and Hessian essentially matter to optimization and generalization of deep learning (Li et al., 2020; Ghorbani et al., 2019; Zhao et al., 2019; Jacot et al., 2019; Yao et al., 2018; Dauphin et al., 2014; Byrd et al., 2011).

A conventional belief is that the covariance is approximately proportional to the Hessian near minima, namely $C(\theta) \propto H(\theta)$ (Jastrzkebski et al., 2017; Zhu et al., 2019; Xie et al., 2020; 2022b; Daneshmand

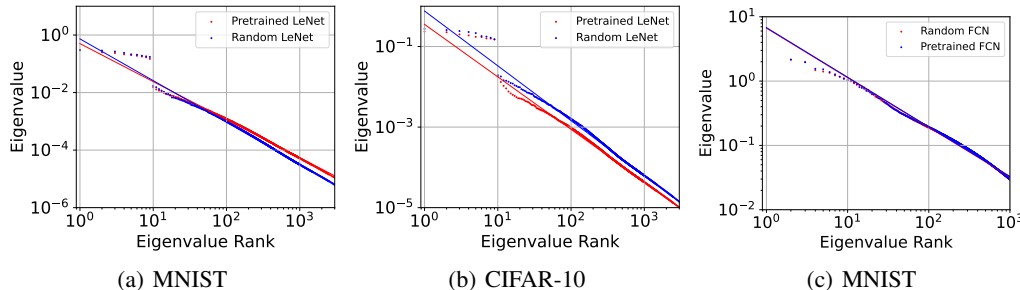

(a) MNIST            (b) CIFAR-10            (c) MNIST

Figure 4: The gradient spectra are highly similar and exhibit power laws for both random models and pretrained models. Model: LeNet and 2-Layer FCN. Dataset: MNIST and CIFAR-10.

Table 2: The KS statistics of the covariance spectra of LeNet and FCN.

| Dataset | Model | Training | $d_{\text{ks}}$ | $d_{\text{c}}$ | Power-Law | $\hat{s}$ |
|---|---|---|---|---|---|---|
| MNIST | LeNet | Pretrain | 0.0206 | 0.0430 | Yes | 1.302 |
| MNIST | LeNet | Random | 0.0220 | 0.0430 | Yes | 1.428 |
| CIFAR-10 | LeNet | Pretrain | 0.0201 | 0.0430 | Yes | 1.257 |
| CIFAR-10 | LeNet | Random | 0.0214 | 0.0430 | Yes | 1.300 |
| MNIST | FCN | Pretrain | 0.0415 | 0.0430 | Yes | 0.866 |
| MNIST | FCN | Random | 0.0418 | 0.0430 | Yes | 0.864 |

et al., 2018; Liu et al., 2021). Mathematically, we have

$$C(\theta) \approx \frac{1}{B}\left[\frac{1}{N}\sum_{j=1}^{N}\nabla l(\theta,(x_j,y_j))\nabla l(\theta,(x_j,y_j))^{\top}\right] = \frac{1}{B}\text{FIM}(\theta) \approx \frac{1}{B}[H(\theta)] \qquad (3)$$

near a critical point, where $\text{FIM}(\theta)$ is the observed Fisher Information matrix, referring to Chapter 8 of Pawitan (2001) and Zhu et al. (2019). The first approximation holds when the expected gradient is small near minima, and the second approximation hold because FIM is approximately equal to the Hessian near minima. Some works (Xie et al., 2020; 2022b) empirically verified Equation (3) and further argued that Equation (3) approximately holds even for random models (which are far from minima) in terms of the flat directions corresponding to small eigenvalues of the Hessian. Note that the most eigenvalues of the Hessian are nearly zero. The gradients along these flat directions are nearly zero as the approximation in Equation (3) is particularly mild along these directions.

The common PCA method, as well as the related low-rank matrix approximation, actually prefers to remove or ignore the components corresponding to small eigenvalues. Because the top eigenvalues and their corresponding eigenvectors can reflect the main properties of a matrix. Unfortunately, previous works (Xie et al., 2020; 2022b) only empirically studied the small eigenvalues of the covariance and the Hessian and missed the most important top eigenvalues. The missing evidence for verifying the top eigenvalues of the covariance and the Hessian can be a serious flaw for the well-known approximation Equation (3). In this paper, we particularly compute the top thousands of eigenvalues of the Hessian and compare them to the corresponding top eigenvalues of the covariance.

In Figure 5, we surprisingly discover that the top eigenvalues of the covariance can significantly deviate from the corresponding eigenvalues of the Hessian sometimes by more than one order of magnitude near or far from minima. Our finding directly challenged the conventional belief on the proportional relation between the covariance and the Hessian near minima.

We also note that the covariance and the second-moment matrix have highly similar spectra in the log-scale plots. For simplicity of expressions, when we say the spectra of gradient noise/gradients in the following analysis, we mean the spectra of the covariance/the second moment, respectively.

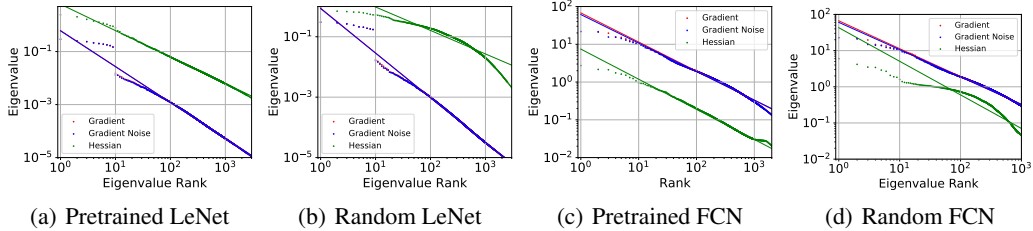

(a) Pretrained LeNet     (b) Random LeNet     (c) Pretrained FCN     (d) Random FCN

Figure 5: The spectra of gradients (the second moment), gradient noise (the covariance), and Hessians for random models and pretrained models. Model: LeNet and FCN. Dataset: MNIST.

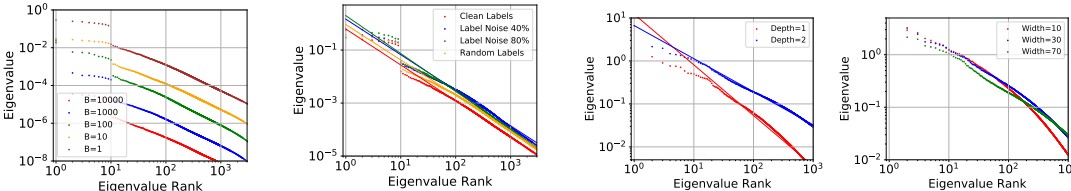

Figure 6: Power-law covariance is approximately inverse to the batch size in deep learning. Model: LeNet. Dataset: MNIST.

Figure 7: Comparison of the gradient spectra on MNIST with clean labels, noisy labels, and random labels. Model: LeNet.

Figure 8: Large enough width (e.g., Width≥ 70) matters to the goodness of power-law covariance, while the depth does not. Left: FCN with various depths. Right: FCN with various widths.

For pretrained models, especially pretrained FCN, while the magnitudes of the Hessian and the corresponding covariance are not even close, the straight lines fit the Hessian spectra and the covariance spectra well. Moreover, the fitted straight lines have similar slopes. Our results also support a very recent finding (Xie et al., 2022a) that the Hessians have power-law spectra for well-trained DNNs but significantly deviate from power laws for random DNNs.

For random models, while the Hessian spectra are not power-law, the covariance spectra surprisingly still exhibit power-law distributions. This is beyond the existing work expected. It is not surprising that the Hessian and the covariance have no close relation without pretraining. However, we report that the power-law covariance spectrum seems to be a universal property, and it is more general than the power-law Hessian spectrum for DNNs.

## 5 EMPIRICAL ANALYSIS AND DISCUSSION

In this section, we empirically studied the covariance spectrum for DNNs in extensive experiments. We particularly reveal that when the power-law covariance for DNNs appears or disappears. As the covariance structure of SGN essentially matters to both optimization and generalization, it will be very interesting to explore how the power-law covariance affects the training trajectories and the Hessian at the learned minima in future work.

**Model:** LeNet (LeCun et al., 1998), Fully Connected Networks (FCN), and ResNet18 (He et al., 2016). **Dataset**: MNIST (LeCun, 1998), CIFAR-10/100 (Krizhevsky & Hinton, 2009), and Avila (De Stefano et al., 2018). Avila is a non-image dataset. We leave the details in Appendix A

**1. Batch Size.** Figure 6 shows that the power-law covariance exists in deep learning for various batch sizes. Moreover, the top eigenvalues are indeed approximately inverse to the batch size as Equation (3) suggests, while the proportional relation between the Hessian and the covariance is very weak.

**2. Learning with Noisy Labels and Random Labels.** Recently, people usually regarded learning with noisy labels (Han et al., 2020) as an important setting for exploring the overfitting and generalization of DNNs. Previous papers (Martin & Mahoney, 2017; Han et al., 2018) demonstrated that

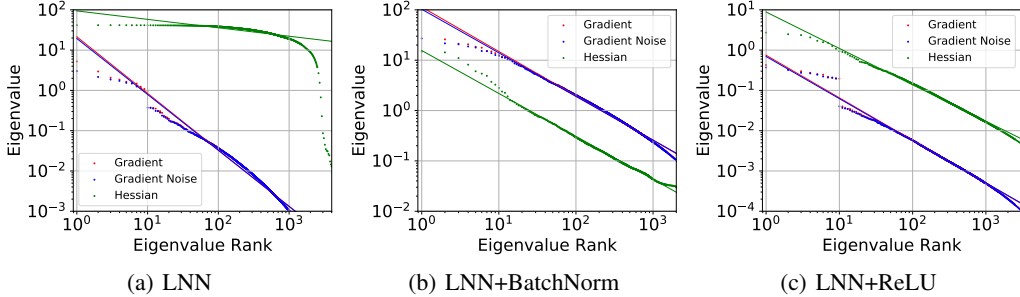

(a) LNN

(b) LNN+BatchNorm

(c) LNN+ReLU

Figure 9: The power-law gradients appear in LNNs with BatchNorm or ReLU, but disappear in fully LNNs. Dataset: MNIST.

DNNs may easily overfit noisy labels and even have completely random labels during training, while convergence speed is slower compared with learning with clean labels. Is this caused by the structure of stochastic gradients? It seems no. We compared the covariance spectrum under clean labels, $40\%$ noisy labels, $80\%$ noisy labels, and completely random labels in Figure 7. We surprisingly discovered that memorization of noisy labels matters little to the power-law structure of stochastic gradients.

**3. Depth and Width.** In this paper, we also study how the depth and the width of neural networks affect the power-law covariance. Figure 8 and the KS tests in Table 7 of Appendix C support that certain width (e.g., Width$\geq 70$) is often required for supporting the power-law hypothesis, while the depth seems unnecessary. Even one-layer FCN may still exhibit the power-law covariance.

**4. Linear Neural Networks (LNNs) and Nonlinearity.** What is the simplest model that shows the power-law covariance? We empirically study the covariance spectra for fully LNNs, LNNs with BatchNorm, and LNNs with ReLU (FCN without BatchNorm) in Figure 9. Obviously, fully LNNs may not learn minima with power-law Hessians. Layer-wise nonlinearity seems necessary for the power-law Hessian spectra (Xie et al., 2022a). However, even the simplest two-layer LNN with no nonlinearity can still exhibit power-law covariance spectra.

**5. The Outliers, BatchNorm, and Data Classes.** We also report that there sometimes exist a few top covariance eigenvalues that significantly deviate from power laws or the fitted straight lines. Figure 10 shows that, the outliers are especially significant for LeNet, a Convolution Neural Network, but less significant for FCN. We also note that LeNet does not apply BatchNorm, while the used FCN applies BatchNorm. What is the real factor that determines whether top outliers are significant or not? Figures 9 and 11 support that it is BatchNorm that makes top outliers less significant rather than the convolution layers. Because even the simplest LNNs, which have no convolution layers and nonlinear activations, still exhibit significant top outliers. This may indicate a novel role of BatchNorm in the training of DNNs.

Suppose there are $c$ classes in the dataset, where $c = 10$ for CIFAR-10 and MNIST. We observe that the number of outliers is usually $c - 1$ in Figures 10 and 12. It supports that the gradients of DNNs indeed usually concentrate in a tiny top space as previous work suggested (Gur-Ari et al., 2018), because the ninth eigenvalue may be larger than the tenth eigenvalue by one order of magnitude. However, this conclusion may not hold similarly well for DNNs without BatchNorm.

Is it possible that the number of outliers depends on the number of model outputs (logits) rather than the number of data classes? In Figure 12, we eliminate the possibility by training a LeNet with 100 logits on CIFAR-10, denoted by CIFAR-10$^\star$. The number of outliers will be constant even if we increase the model logits.

**6. Optimization.** In Figure 13, we discover that Weight Decay and Momentum do not affect the power-law structure, while Adam obviously breaks power laws due to adaptive gradients.

**7. Non-image Data.** It is known that natural images have some special statistical properties (Torralba & Oliva, 2003). May the power-law covariance be caused by the statistical properties of natural images. In Figure 14, we conduct the experiment on a classical non-image UCI dataset, Avila, which

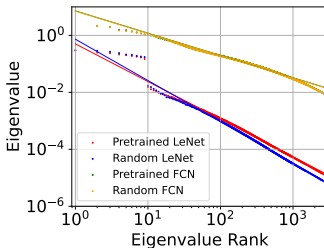 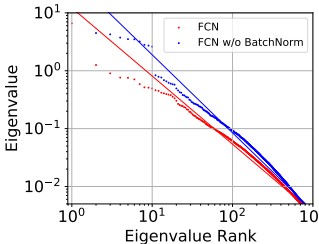 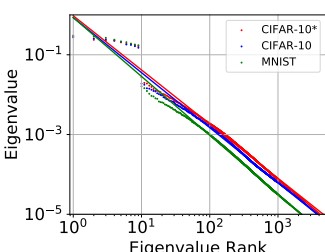

Figure 10: The number of outliers is usually $c-1$. The outliers of the FCN gradient spectrum are much less significant than that of LeNet. Dataset: MNIST.

Figure 11: BatchNorm can make the outliers less significant. Model: FCN with/without BatchNorm.

Figure 12: The outliers in the power-law spectrum mainly depends on the number of data classes rather than the number of model outputs (logits).

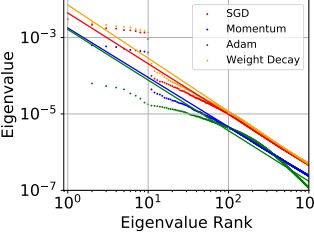 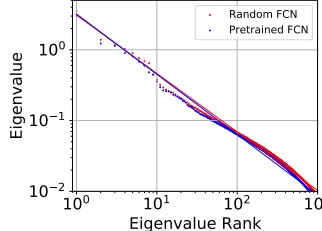 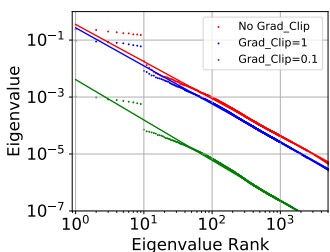

Figure 13: The gradient spectra with various optimization techniques. Dataset: CIFAR-10. Model: LeNet.

Figure 14: The power-law gradients appear on non-image natural datasets. Dataset: Avila. Model: FCN.

Figure 15: The power-law covariance holds with Gradient Clipping. Model: LeNet. Dataset: CIFAR-10.

is a simple classification dataset with only ten input features. The existence of the power-law gradients of DNNs seems more general than natural image statistics.

**8. Gradient Clipping.** Gradient Clipping is a popular method for stabilizing and accelerating the training of language models (Zhang et al., 2019). Figure 15 shows that Gradient Clipping does not break the power-law covariance structure.

**9. Limitations.** Our work does not theoretically touch on why the power-law covariance generally exists in deep learning. The theoretical mechanism behind the elegant mathematical structure may be promising for understanding deep learning. We will leave it as future work.

## 6 CONCLUSION

In this paper, we revisit two essentially important topics on stochastic gradients in deep learning with extensive empirical results and formal statistical evidences. First, we reconciled recent conflicting arguments on the heavy-tail properties of SGN. We demonstrated that dimension-wise gradients usually have power-law heavy tails, while iteration-wise gradients or SGN have relatively high Gaussianity. Second, to our knowledge, we are the first to report that the covariance/the second moment of gradients usually has a power-law structure for various neural networks. The heavy tails of dimension-wise gradients could be explained as a natural result of the power-law covariance. We further analyze how various settings affect the power-law covariance structure in deep learning. Our work not only provides rich insights into the structure of stochastic gradients, but also may point to novel approaches to understanding stochastic optimization for deep learning in the future.

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

## A  EXPERIMENTAL SETTINGS

**Computational environment.** The experiments are conducted on a computing cluster with NVIDIA® V100 GPUs and Intel® Xeon® CPUs.

### A.1  GRADIENT HISTORY MATRICES

In this paper, we compute the Gradient History Matrices and the covariance for multiple models on multiple datasets. Then, we use the elements in Gradient History Matrices and the eigenvalues of the covariance/second-moment to evaluate the goodness of fitting Gaussian distributions or power-law distributions via $\chi^2$ tests and KS tests.

The Gradient History Matrix is an $n \times T$ matrix. For the experiment of LeNet and FCN, we compute the gradients for $T = 5000$ iterations at a fixed randomly initialized position $\theta^{(0)}$ or a pretrained position $\theta^\star$. Due to limit of memory capacity, for the experiment of ResNet18, we compute the gradients for $T = 200$ iterations at $\theta^{(0)}$ or $\theta^\star$.

A Gradient History Matrix can be used to compute the covariance or the second moment of stochastic gradients for a neural network. Note that a covariance matrix is an $n \times n$ matrix, which is extremely large for modern neural networks. Thus, we mainly analyze the gradient structures of LeNet and FCN at an affordable computational cost.

### A.2  MODELS AND DATASETS

**Models:** LeNet (LeCun et al., 1998), Fully Connected Networks (FCN), and ResNet18 (He et al., 2016). We mainly used two-layer FCN which has 70 neurons for each hidden layer, ReLu activations, and BatchNorm layers, unless we specify otherwise.

**Datasets:** MNIST (LeCun, 1998) and CIFAR-10/100 (Krizhevsky & Hinton, 2009), and non-image Avila (De Stefano et al., 2018).

**Optimizers:** SGD, SGD with Momentum, and Adam (Kingma & Ba, 2015).

### A.3  IMAGE CLASSIFICATION ON MNIST

We perform the common per-pixel zero-mean unit-variance normalization as data preprocessing for MNIST.

**Pretraining Hyperparameter Settings:** We train neural networks for 50 epochs on MNIST for obtaining pretrained models. For the learning rate schedule, the learning rate is divided by 10 at the epoch of $40\%$ and $80\%$. We use $\eta = 0.1$ for SGD/Momentum and $\eta = 0.001$ for Adam. The batch size is set to 128. The strength of weight decay defaults to $\lambda = 0.0005$ for pretrained models. We set the momentum hyperparameter $\beta_1 = 0.9$ for SGD Momentum. As for other optimizer hyperparameters, we apply the default settings directly.

**Hyperparameter Settings for $G$:** We use $\eta = 0.1$ for SGD/Momentum and $\eta = 0.001$ for Adam. The batch size is set to 1 and no weight decay is used, unless we specify them otherwise.

### A.4 IMAGE CLASSIFICATION ON CIFAR-10 AND CIFAR-100

**Data Preprocessing For CIFAR-10 and CIFAR-100:** We perform the common per-pixel zero-mean unit-variance normalization, horizontal random flip, and $32 \times 32$ random crops after padding with 4 pixels on each side.

**Pretraining Hyperparameter Settings:** In the experiments on CIFAR-10 and CIFAR-100: $\eta = 1$ for Vanilla SGD; $\eta = 0.1$ for SGD (with Momentum); $\eta = 0.001$ for Adam. For the learning rate schedule, the learning rate is divided by 10 at the epoch of $\{80, 160\}$ for CIFAR-10 and $\{100, 150\}$ for CIFAR-100, respectively. The batch size is set to 128 for both CIFAR-10 and CIFAR-100. The batch size is set to 128 for MNIST, unless we specify it otherwise. The strength of weight decay defaults to $\lambda = 0.0005$ as the baseline for all optimizers unless we specify it otherwise. We set the momentum hyperparameter $\beta_1 = 0.9$ for SGD and adaptive gradient methods which involve in Momentum. As for other optimizer hyperparameters, we apply the default settings directly.

**Hyperparameter Settings for $G$:** We use $\eta = 1$ for SGD, $\eta = 0.1$ for SGD with Momentum, and $\eta = 0.001$ for Adam. The batch size is set to 1 and no weight decay is used, unless we specify them otherwise.

### A.5 LEARNING WITH NOISY LABELS

We trained LeNet via SGD (with Momentum) on corrupted MNIST with various (asymmetric) label noise. We followed the setting of Han et al. (2018) for generating noisy labels for MNIST. The symmetric label noise is generated by flipping every label to other labels with uniform flip rates $\{40\%, 80\%\}$. In this paper, we used symmetric label noise.

For obtaining datasets with random labels which have little knowledge behind the pairs of instances and labels, we also randomly shuffle the labels of MNIST to produce Random MNIST.

## B GOODNESS-OF-FIT TESTS

### B.1 KOLMOGOROV-SMIRNOV TEST

In this section, we introduce how to conduct the Kolmogorov-Smirnov Goodness-of-Fit Test.

We used Maximum Likelihood Estimation (MLE) (Myung, 2003; Clauset et al., 2009) for estimating the parameter $\beta$ of the fitted power-law distributions and the Kolmogorov-Smirnov Test (KS Test) (Massey Jr, 1951; Goldstein et al., 2004) for statistically testing the goodness of fitting power-law distributions. The KS test statistic is the KS distance $d_{\mathrm{ks}}$ between the hypothesized (fitted) distribution and the empirical data, which measures the goodness of fit. It is defined as

$$d_{\mathrm{ks}} = \sup_{\lambda} |F^{\star}(\lambda) - \hat{F}(\lambda)|, \tag{4}$$

where $F^{\star}(\lambda)$ is the hypothesized cumulative distribution function and $\hat{F}(\lambda)$ is the empirical cumulative distribution function based on the sampled data (Goldstein et al., 2004). The estimated power exponent via MLE (Clauset et al., 2009) can be written as

$$\hat{\beta} = 1 + K \left[ \sum_{i=1}^{K} \ln \left( \frac{\lambda_i}{\lambda_{\min}} \right) \right]^{-1}, \tag{5}$$

where $K$ is the number of tested samples and we set $\lambda_{\min} = \lambda_k$. In this paper, we choose the top $K = 1000$ data points for the power-law hypothesis tests, unless we specify it otherwise. We note that the Powerlaw library (Alstott et al., 2014) provides a convenient tool to compute the KS distance, $d_{\mathrm{ks}}$, and estimate the power exponent.

According to the practice of Kolmogorov-Smirnov Test (Massey Jr, 1951), we state ***the null hypothesis*** that the tested spectrum is not power-law. We state the alternative hypothesis, called ***the***

**power-law hypothesis**, that the tested spectrum is power-law. If $d_{\mathrm{ks}}$ is higher than the critical value $d_{\mathrm{c}}$ at the $\alpha = 0.05$ significance level, we would accept the null hypothesis. In contrast, if $d_{\mathrm{ks}}$ is lower than the critical value $d_{\mathrm{c}}$ at the $\alpha = 0.05$ significance level, we would reject the null hypothesis and accept the power-law hypothesis.

For each KS test in this paper, we select top $k = 1000$ data points from dimension-wise gradients and iteration-wise gradients and top $k = 1000$ covariance eigenvalues as the tested sets to measure the goodness of power laws. We choose the largest data points for two reasons. First, focusing on relatively large values is very reasonable and common in various fields' power-law studies (Stringer et al., 2019; Reuveni et al., 2008; Tang & Kaneko, 2020), as real-world distributions typically follow power laws only after/large than some cutoff values (Clauset et al., 2009) for ensuring the convergence of the probability distribution. Second, researchers are usually more interested in significantly large eigenvalues due to the low-rank matrix approximation.

### B.2 $\chi^2$ TEST

In this section, we introduce how we conduct $\chi^2$ Test to evaluate the Gaussianity.

We directly used the $\chi^2$ Normal Test implemented by the classical Python-based scientific computing package, Scipy (Virtanen et al., 2020), to evaluate the Gaussianity of empirical data. Note that we need to normalize the empirical data via whitening (zero-mean and unit-variance) before the tests.

The Gaussianity test statistic, $p$-value, is returned by the squared sum of the statistics of Skewness Test and Kurtosis Test (Cain et al., 2017). Skewness is a measure of symmetry. A distribution or data set is symmetric if it looks the same to the left and right of the center. Kurtosis is a measure of whether the data are heavy-tailed or light-tailed relative to a normal distribution. Empirical data with high kurtosis tend to have heavy tails. Empirical data with low kurtosis tend to have light tails. Thus, $\chi^2$ Test can reflect both Gaussianity and heavy tails. In this paper, we randomly choose $K = 100$ data points for the Gaussian hypothesis tests, unless we specify it otherwise.

We may write the $p$-value return by $\chi^2$ Test as

$$p = z_S^2 + z_K^2, \tag{6}$$

where $z_S$ is the Skewness Test statistic and $z_K$ is the Kurtosis Test statistic. There are a number of ways to compute $z_S$ and $z_K$ in practice. It is convenient to use the default two-sided setting in Virtanen et al. (2020). Please refer to Virtanen et al. (2020) and the source code of $stats.skewtest$ and $stats.kurtosistest$ for the detailed implementation.

For each $\chi^2$ test in this paper, we randomly select $k = 100$ data points from both dimension-wise gradients and iteration-wise gradients as the tested set to measure the Gaussianity. The returned test statistic, $p$-value, is a classical indicator of the relative goodness of Gaussianity for two types of gradients.

## C   STATISTICAL TEST RESULTS

We present the statistical test results of dimension-wise gradients and iteration-wise gradients of LeNet and ResNet18 on various datasets in Tables 3 and 4.

We conducted the KS Tests for all of our studied covariance spectra. We display the KS test statistics and the estimated power exponents $\hat{s}$ in the tables. For better visualization, we color accepting the power-law hypothesis in blue and color accepting the null hypothesis (and the cause) in red. The KS Test statistics of the covariance spectra are shown in Tables 5, 6, 7, 8, 9, and 10.

Table 3: The KS and $\chi^2$ statistics and the hypothesis acceptance rates of iteration-wise gradients with respect to the batch size. Model: LeNet. Dataset: MNIST

| Type | Training | Setting | $\bar{d}_{ks}$ | $d_c$ | Power-Law Rate | $\bar{p}$-value | Gaussian Rate |
|------|----------|---------|------|------|----------------|---------|---------------|
| Iteration | Random | $B = 1$ | 0.428 | 0.0430 | 0.067% | 0.047 | 12.6% |
| Iteration | Random | $B = 3$ | 0.385 | 0.0430 | 0.17% | 0.089 | 21.5% |
| Iteration | Random | $B = 10$ | 0.267 | 0.0430 | 0.25% | 0.173 | 28.5% |
| Iteration | Random | $B = 30$ | 0.249 | 0.0430 | 0.16% | 0.240 | 50.9% |
| Iteration | Random | $B = 100$ | 0.191 | 0.0430 | 0.079% | 0.321 | 65.1% |
| Iteration | Random | $B = 300$ | 0.119 | 0.0430 | 0.033% | 0.382 | 74.5% |
| Iteration | Random | $B = 1000$ | 0.120 | 0.0430 | 0.041% | 0.388 | 75.5% |
| Dimension | Random | $B = 1$ | 0.0306 | 0.0430 | 90.6% | $4.51 \times 10^{-5}$ | 0% |
| Dimension | Random | $B = 3$ | 0.0358 | 0.0430 | 74.5% | $9.07 \times 10^{-5}$ | 0.02% |
| Dimension | Random | $B = 10$ | 0.0392 | 0.0430 | 65.3% | $1.78 \times 10^{-4}$ | 0% |
| Dimension | Random | $B = 30$ | 0.0379 | 0.0430 | 68.9% | $2.29 \times 10^{-4}$ | 0.14% |
| Dimension | Random | $B = 100$ | 0.0355 | 0.0430 | 76.6% | $4.11 \times 10^{-4}$ | 0.18% |
| Dimension | Random | $B = 300$ | 0.0269 | 0.0430 | 97.5% | $1.21 \times 10^{-3}$ | 0.48% |
| Dimension | Random | $B = 1000$ | 0.0309 | 0.0430 | 90.6% | $1.43 \times 10^{-4}$ | 0% |

Table 4: The KS and $\chi^2$ statistics and the hypothesis acceptance rates of the gradients over dimensions and iterations, respectively. Model: ResNet18. Batch Size: 100.

| Dataset | Training | SG Type | $\bar{d}_{ks}$ | $d_c$ | Power-Law Rate | $\bar{p}$-value | Gaussian Rate |
|---------|----------|---------|------|------|----------------|---------|---------------|
| CIFAR-10 | Random | Dimension | 0.0924 | 0.0962 | 54.3% | $1.73 \times 10^{-2}$ | 6.4% |
| CIFAR-10 | Random | Iteration | 0.141 | 0.0962 | 1.32% | 0.495 | 93.4% |
| CIFAR-10 | Pretrain | Dimension | 0.0717 | 0.0962 | 82.6% | $1.1 \times 10^{-2}$ | 3.2% |
| CIFAR-10 | Pretrain | Iteration | 0.140 | 0.0962 | 1.38% | 0.497 | 93.5% |
| CIFAR-100 | Random | Dimension | 0.0631 | 0.0962 | 92.4% | $8.55 \times 10^{-3}$ | 3% |
| CIFAR-100 | Random | Iteration | 0.141 | 0.0962 | 1.36% | 0.496 | 93.2% |
| CIFAR-100 | Pretrain | Dimension | 0.0637 | 0.0962 | 88.5% | $8.11 \times 10^{-3}$ | 3.4% |
| CIFAR-100 | Pretrain | Iteration | 0.140 | 0.0962 | 1.37% | 0.496 | 93.1% |

Table 5: The KS statistics of the second-moment spectra of dimension-wise gradients for LeNet on MNIST.

| Dataset | Model | Training | Batch | Sample size | Setting | $d_{ks}$ | $d_c$ | Power-Law | $\hat{s}$ |
|---------|-------|----------|-------|-------------|---------|------|------|-----------|-----------|
| MNIST | LeNet | Pretrain | 1 | 1000 | - | 0.0206 | 0.0430 | Yes | 1.302 |
| MNIST | LeNet | Pretrain | 10 | 1000 | - | 0.0244 | 0.0430 | Yes | 1.313 |
| MNIST | LeNet | Pretrain | 100 | 1000 | - | 0.0171 | 0.0430 | Yes | 1.390 |
| MNIST | LeNet | Pretrain | 1000 | 1000 | - | 0.0173 | 0.0430 | Yes | 1.314 |
| MNIST | LeNet | Pretrain | 10000 | 1000 | - | 0.0204 | 0.0430 | Yes | 1.290 |
| MNIST | LeNet | Pretrain | 60000 | 1000 | - | 0.106 | 0.0430 | No | 0.206 |
| MNIST | LeNet | Random | 1 | 1000 | - | 0.0220 | 0.0430 | Yes | 1.428 |
| MNIST | LeNet | Random | 10 | 1000 | - | 0.0223 | 0.0430 | Yes | 1.334 |
| MNIST | LeNet | Random | 100 | 1000 | - | 0.0228 | 0.0430 | Yes | 1.313 |
| MNIST | LeNet | Random | 1000 | 1000 | - | 0.0198 | 0.0430 | Yes | 1.423 |
| MNIST | LeNet | Random | 10000 | 1000 | - | 0.0213 | 0.0430 | Yes | 1.284 |
| MNIST | LeNet | Random | 60000 | 1000 | - | 0.203 | 0.0430 | No | 0.271 |

Table 6: The KS statistics of the covariance spectra of dimension-wise gradients for LeNet on MNIST.

| Dataset | Model | Training | Batch | Sample size | Setting | $d_{\mathrm{ks}}$ | $d_{\mathrm{c}}$ | Power-Law | $\hat{s}$ |
|---------|-------|----------|-------|-------------|---------|-------|-------|-----------|-----------|
| MNIST | LeNet | Random | 1 | 1000 | - | 0.0226 | 0.0430 | Yes | 1.425 |
| MNIST | LeNet | Random | 10 | 1000 | - | 0.0227 | 0.0430 | Yes | 1.331 |
| MNIST | LeNet | Random | 100 | 1000 | - | 0.0230 | 0.0430 | Yes | 1.311 |
| MNIST | LeNet | Random | 1000 | 1000 | - | 0.0200 | 0.0430 | Yes | 1.423 |
| MNIST | LeNet | Random | 10000 | 1000 | - | 0.0287 | 0.0430 | Yes | 1.320 |
| MNIST | LeNet | Pretrain | 1 | 1000 | - | 0.0206 | 0.0430 | Yes | 1.299 |
| MNIST | LeNet | Pretrain | 10 | 1000 | - | 0.0247 | 0.0430 | Yes | 1.310 |
| MNIST | LeNet | Pretrain | 100 | 1000 | - | 0.0171 | 0.0430 | Yes | 1.386 |
| MNIST | LeNet | Pretrain | 1000 | 1000 | - | 0.0174 | 0.0430 | Yes | 1.312 |
| MNIST | LeNet | Pretrain | 10000 | 1000 | - | 0.0223 | 0.0430 | Yes | 1.331 |
| MNIST | LeNet | Pretrain | 1 | 1000 | Label Noise 40% | 0.0289 | 0.0430 | Yes | 1.453 |
| MNIST | LeNet | Pretrain | 1 | 1000 | Label Noise 80% | 0.0138 | 0.0430 | Yes | 11.442 |
| MNIST | LeNet | Pretrain | 1 | 1000 | Random Label | 0.0129 | 0.0430 | Yes | 1.374 |
| MNIST | LeNet | Pretrain | 1 | 1000 | GradClip=1 | 0.0226 | 0.0430 | Yes | 1.323 |
| MNIST | LeNet | Pretrain | 1 | 1000 | GradClip=0.1 | 0.0261 | 0.0430 | Yes | 1.343 |

Table 7: The KS statistics of the second-moment spectra of dimension-wise gradients for FCN on MNIST.

| Dataset | Model | Training | Batch | Sample size | Setting | $d_{\mathrm{ks}}$ | $d_{\mathrm{c}}$ | Power-Law | $\hat{s}$ |
|---------|-------|----------|-------|-------------|---------|-------|-------|-----------|-----------|
| MNIST | 2Layer-FCN | Pretrain | 10 | 1000 | - | 0.0415 | 0.0430 | Yes | 0.866 |
| MNIST | 2Layer-FCN | Random | 10 | 1000 | - | 0.0418 | 0.0430 | Yes | 0.864 |
| MNIST | 2Layer-FCN | Random | 10 | 1000 | Noise | 0.0427 | 0.0430 | Yes | 0.862 |
| MNIST | 2Layer-FCN | Pretrain | 10 | 1000 | Width=70 | 0.0415 | 0.0430 | Yes | 0.866 |
| MNIST | 2Layer-FCN | Pretrain | 10 | 1000 | Width=30 | 0.0425 | 0.0430 | Yes | 0.869 |
| MNIST | 2Layer-FCN | Pretrain | 10 | 1000 | Width=10 | 0.0486 | 0.0430 | No | |
| MNIST | 2Layer-FCN | Random | 10 | 1000 | Width=70 | 0.0418 | 0.0430 | Yes | 0.864 |
| MNIST | 2Layer-FCN | Random | 10 | 1000 | Width=30 | 0.0488 | 0.0430 | No | |
| MNIST | 2Layer-FCN | Random | 10 | 1000 | Width=10 | 0.0491 | 0.0430 | No | |
| MNIST | 1Layer-FCN | Random | 10 | 1000 | - | 0.0384 | 0.0430 | Yes | 1.357 |
| MNIST | 1Layer-FCN | Pretrain | 10 | 1000 | - | 0.0384 | 0.0430 | Yes | 1.355 |

Table 8: The KS statistics of the second-moment spectra of dimension-wise gradients for LNN on MNIST.

| Dataset | Model | Training | Batch | Sample size | Setting | $d_{\mathrm{ks}}$ | $d_{\mathrm{c}}$ | Power-Law | $\hat{s}$ |
|---------|-------|----------|-------|-------------|---------|-------|-------|-----------|-----------|
| MNIST | 4Layer-LNN | Pretrain | 10 | 1000 | - | 0.0445 | 0.0430 | No | 1.629 |
| MNIST | 4Layer-LNN | Pretrain | 10 | 1000 | BatchNorm | 0.0268 | 0.0430 | Yes | 0.955 |
| MNIST | 4Layer-LNN | Pretrain | 10 | 1000 | ReLU | 0.0154 | 0.0430 | Yes | 1.074 |

Table 9: The KS statistics of the covariance spectra of dimension-wise gradients for LeNet on CIFAR-10.

| Dataset | Model | Training | Batch | Sample size | Setting | $d_{\mathrm{ks}}$ | $d_{\mathrm{c}}$ | Power-Law | $\hat{s}$ |
|---------|-------|----------|-------|-------------|---------|-------|-------|-----------|-----------|
| CIFAR-10 | LeNet | Pretrain | 1 | 1000 | - | 0.0201 | 0.0430 | Yes | 1.257 |
| CIFAR-10 | LeNet | Random | 1 | 1000 | - | 0.0214 | 0.0430 | Yes | 1.300 |
| CIFAR-10 | LeNet | Pretrain | 1 | 1000 | GradClip=0.1 | 0.0244 | 0.0430 | Yes | 1.348 |
| CIFAR-10 | LeNet | Random | 100 | 1000 | SGD | 0.00818 | 0.0430 | Yes | 1.305 |
| CIFAR-10 | LeNet | Random | 100 | 1000 | Weight Decay | 0.0107 | 0.0430 | Yes | 1.300 |
| CIFAR-10 | LeNet | Random | 100 | 1000 | Momentum | 0.00806 | 0.0430 | Yes | 1.262 |
| CIFAR-10 | LeNet | Random | 100 | 1000 | Adam | 0.0634 | 0.0430 | No | |

Table 10: The KS statistics of the covariance spectra of LeNet on CIFAR-100.

| Dataset | Model | Training | Batch | Sample size | Setting | $d_{\mathrm{ks}}$ | $d_{\mathrm{c}}$ | Power-Law | $\hat{s}$ |
|---------|-------|----------|-------|-------------|---------|-------------------|------------------|-----------|-----------|
| CIFAR-100 | LeNet | Pretrain | 1 | 1000 | - | 0.0287 | 0.0430 | Yes | 1.276 |
| CIFAR-100 | LeNet | Random | 1 | 1000 | - | 0.0307 | 0.0430 | Yes | 1.229 |
| CIFAR-100 | LeNet | Pretrain | 1 | 1000 | - | 0.0197 | 0.0430 | Yes | 1.076 |

