# OpenReview forum: "Rethinking the Structure of Stochastic Gradients: Empirical and Statistical Evidence"
_ICLR.cc/2023/Conference — Submitted to ICLR 2023_

### Official Review · Reviewer_cbEm · 2022-10-19

**Confidence:** 4
**Correctness:** 3
**Technical Novelty And Significance:** 2
**Empirical Novelty And Significance:** 3
**Recommendation:** 5

**Clarity, Quality, Novelty And Reproducibility:**

The quality and reproducibility seem good.

As for the novelty, perhaps the authors could emphasize the advances with respect to the works belonging to Research Line 2 (Xie et al 2020, Xie et al 2022b, Li et al 2019).

Though the paper seems generally well-written, there are many places where I am not sure about what was done or about what a specific message meant. This makes it harder for me to assess the paper.

1) Section 2, when defining the KS test, the authors say:
"If the KS distance $d_{ks}$ is smaller than the critical distance $d_c$ , the KS test will reject the power-law hypothesis."
Immediately after I read:
"The smaller $d_{ks}$ is, the better the goodness-of-power-law is."
This is confusing.

2) Section 2, when defining the $\chi^2$ test.
"If the estimated $p$-value is less than 0.05, the $\chi^2$ test will reject the Gaussian hypothesis."
Immediately after I read:
"The smaller $p$-value is, the better the goodness-of-Gaussianity is."
This is confusing too.

3) I am not sure about Figure 1 (and others). The dimension-wise gradients should have n components (the # of weights). Since the model is a LeNet on cifar-10, I would expect there to be 62000 points on the x axis, but there are only 2000. As for the iteration-wise gradients, the run has 5000 iterations (App.A), so again it looks like not the full curves are shown.

4) In Fig.1 it is not clear at what time the Dimension-wise curve is taken. Given that there are so few points, I don't think it comes from more than one time. Is it at initialization?

5) In Fig.2 there is a red curve for the power law rate on the legend, but the curve is not visible. Is it at y=0? I would make it visible.

6) Some times the main text is not self explanatory, and it is necessary to go to the appendix to understand the tables/figures. For example, column 2 of table 1 (Training) is not explained in the main text.

7) Though the names might sound intuitive, I often got confused between the two kinds of noise. I would suggest a table similar to the following:

------------------------------------------------
Iteration-wise     | Fixed component, All the dynamics   | Gaussian
------------------------------------------------
Dimension-wise | All components, Fixed time                | Power law
------------------------------------------------

8) I am not sure I understand why the authors say that the true SGN is the iteration-wise (e.g. "the heavy-tail property describes dimension-wise gradients (not SGN)"). As said in other parts of the text, the heavy-tailed dimension-wise noise (i.e. the fact that at a fixed step the gradient feels some strong kicks toward specific directions) is what is actually interesting for implicit regularization. This is also my understanding of what is treated by Simsekli et al 2019, and if I were to focus on one of the two kinds of noise, it would be dimension-wise.
So I am not sure about the conclusions of section 3, and I am not sure I understand the request to Simsekli and coauthors to clarify this.

9) Figure 3 is not mentioned in Sec.3, but only in the introductory part.

10) Is Fig.4 (and the others) before running the dynamics, or are they near a minimum? Is there a difference in the spectra at the beginning of the dynamics and at initialization? It is hard to interpret the results of sec.4 without being sure about this.

11) It would be great if the figures in page 7 were made a little bit bigger or more readable (e.g. Fig.5, the legend could have bigger dots, and the curves could have more different colors)

12) Also, Eq.3 shows that the batch size is important in the correspondence between the spectra of Hessian and Covariance, but it is not reported in Fig.5. Further, why don't they rescale the eigenvalues by the batch size, since that's the quantity of interest?

13) Why do the authors emphasize that the highest eigenvalues of the hessian are different when looking at Hessian and Covariance? The full spectra seem different (including the power law tails in Fig5a), not only the largest eigenvalues. If the intention is to emphasize that the largest eigenvalues are different, this is already well-established (e.g. arXiv:1812.04754 or arXiv:1706.04454).


**Strength And Weaknesses:**

Strengths:
- The distinction between two different kinds of stochastic gradient noise allows to provide a clearer picture of the phenomenon
- The power law of the covariance spectrum was not observed before and seems to be very stable across different situations


Weaknesses:
- Presentation: Though the paper is well-written, several things were not clear to me (see next section).
- Averaging: Curves don't seem averaged over several instances
- Stages of dynamics: There is no emphasis on different stages of the dynamics, but we know that at least three distinct phases of learning can be identified (see e.g. arXiv:1803.06969 or arXiv:2111.07167). This might as well help explaining why the Gaussian and Power-Law rates are far from 100%. For example, arXiv:1910.09626 notices that there might be differences along training.


**Summary Of The Paper:**

The authors pick up a controversy on the nature of the stochastic gradient noise, and aim at clarifying it by distinguishing among iteration-wise and dimension-wise noise. The iteration-wise noise looks separately at a single component throughout the whole dynamics, and it is more Gaussian distributed. The dimension-wise noise looks at all components at fixed time, and looks more power-law-distributed.
Then, they show empirically that the covariance spectra follow a power law (this seems to hold in different settings), and emphasize the connections/differences between them and the hessian of the loss.



**Summary Of The Review:**

The paper is simple but it passes a couple of ideas worth disseminating. My main concerns are about the presentation, because some of the results were not too clear to me and it was therefore hard to assess them. My second concern is that there often does not seem to be a distinction between different phases of learning.

---

> ### Author Response · Authors · 2022-11-18
> **Responses to Reviewer cbEm**
>
> We highly appreciate Reviewer ebEm’s helpful comments and hard work.
>
> We will revise the paper according to your constructive suggestions.
>
> We kindly argue that your main concern may be addressed.
>
> Q1: Stage of Learning Dynamics.
>
> A1: This is a nice question. We actually studied the gradient structures of both randomly initialized DNNs and pretrained DNNs. As Figures 3, 4, 5 show, the gradient structures of the early-phase models and the final-phase models are quite similar, while the Hessian spectrums of the early-phase models and the final-phase models are significantly different (Please see Figure 5). Note that the scope of this paper focuses on the gradient structures. In our on-going experiment, we evaluate the gradient structure of DDNs per 10 epochs during training, and we still did not see the phase transition of gradient structures with respect to stages of learning dynamics.
>
> Q2: The confusing expression on the KS distance and p-value in Section 2.
>
> A2: Thanks for pointing out the typos. We will correct the statements. If $d_{ks}$/$p$-value is smaller than some bars, we will not reject the power-law/Gaussian hypothesis and the goodness-of-fitting is better.
>
> Q3: It looks like not the full curves are shown.
>
> A3: Yes, because it is reasonable to plot the top eigenvalues or gradients larger than some minimal cutoff value according to the magnitude rank, when people focus on the heavy tails. (See Appendix B.1) This is common in practice for multiple reasons [1,2]. First, focusing on relatively large values is very reasonable and common in various fields' power-law studies, as real-world distributions typically follow power laws only after/large than some cutoff values for ensuring the convergence of the probability distribution. Second, researchers are usually more interested in significantly large eigenvalues/elements which contribute more to the Hessian/the gradient structure. Third, empirically estimating a large number of nearly zero eigenvalues/elements can very inaccurate and noisy.
>
> Q4: In Fig.1 it is not clear at what time the Dimension-wise curve is taken. Is it at initialization?
>
> A4: Yes, Figure 1 plot the curves of a randomly initialized model. We can see more curves of pretrained models and initialized models in Figures 3-5.
>
>
> Q5: I am not sure I understand why the authors say that the true SGN is the iteration-wise (e.g. "the heavy-tail property describes dimension-wise gradients (not SGN)").
>
> A5: Dimension-wise gradient noise can be important for training, but it is iteration-wise gradient noise that explicitly plays a key role in learning dynamics of SGD. In continuous-time dynamics/Langevin Dynamics/SDEs of SGD, we must specify the type of iteration-wise gradient noise which is mainly caused by minibatch training. Simsekli et al 2019 claimed that iteration-wise gradient noise is heavy-tailed, while the empirical evidence only supports that dimension-wise gradient noise is heavy-tailed. A number of papers (Xie et al., 2020; 2022b; Li et al., 2021) pointed out this issue and made some discussion.
>
> Q6: Is Fig.4 (and the others) before running the dynamics, or are they near a minimum? Is there a difference in the spectra at the beginning of the dynamics and at initialization? It is hard to interpret the results of sec.4 without being sure about this.
>
> A6: We plot the curves of both pretrained models and initialized models in Figure 4. We will make this point more clear.
>
> Q7: Eq.3 shows that the batch size is important in the correspondence between the spectra of Hessian and Covariance, but it is not reported in Fig.5. Further, why don't they rescale the eigenvalues by the batch size, since that's the quantity of interest?
>
> A7: We plot the covariance spectra with respect to various batch sizes in Figure 6, which supports Eq.3. In Figure 5, we let the batch size be 1.
>
> Q8: Why do the authors emphasize that the highest eigenvalues of the hessian are different when looking at Hessian and Covariance? The full spectra seem different (including the power law tails in Fig5a), not only the largest eigenvalues.
>
> A8: We agree that the spectra that capture more than thousands of largest eigenvalues are different. Previous papers (Xie et al., 2020; 2022b; Zhu et al. 2019) reported that Hessian and Covariance are highly similar near minima, but only presented empirical evidence of small-magnitude eigenvalues. Our results show that at least top thousands of eigenvalues which follow power laws are very different, beyond the existing belief.
>
> References:
>
> [1] Aaron Clauset, Cosma Rohilla Shalizi, and Mark EJ Newman. Power-law distributions in empirical data. SIAM review, 51(4):661–703, 2009.
>
> [2] Valentin Thomas, Fabian Pedregosa, Bart Merriënboer, Pierre-Antoine Manzagol, Yoshua Bengio, and Nicolas Le Roux. On the interplay between noise and curvature and its effect on optimization and generalization. In International Conference on Artificial Intelligence and Statistics, pp. 3503–3513. PMLR, 2020.

---

> > ### Comment · Reviewer_cbEm · 2022-11-24
> > **Thanks for your answer**
> >
> > I thank the authors for their replies to my comments.
> >
> > The following points still do not convince me completely:
> >
> > R1. I appreciate that the authors study the two cases, since transfer learning is currently very used. Despite the differences I see both considered situations as initializations, since the loss is high, the learning of the last layer is arguably the most important [http://www.cs.utoronto.ca/~rgrosse/cacm2011-cdbn.pdf], and the initial dynamics in transfer learning is essentially driven by the gradients of the last layer if the last layer was not tuned yet (and I think it is nontrivial how this happens and changes with time).
> > Therefore, I believe that the claims should be reduced, since they might not apply to the full learning.
> >
> > R3.
> > I see the point about the fluctuations at the tails (which can be improved by increasing the statistics) and am familiar with exponential cutoffs (which usually scale with the system size, so one can do finite-size scaling), but how are the first 2k out of 62k points significative of the tail of the distribution? I do not see that the plots are sufficient evidence to support the statement.
> >
> > R8. I find it confusing to state that the difference is in the first eigenvalues, when it is on the whole spectrum. I suggest rephrasing.

---

> > > ### Author Response · Authors · 2022-11-24
> > > **Responses to Updated Reviews of Reviewer cbEm**
> > >
> > > We sincerely appreciate the updated reviews of Reviewer cbEm.
> > >
> > > R1: I appreciate that the authors study the two cases, since transfer learning is currently very used. Despite the differences I see both considered situations as initializations, since the loss is high.
> > >
> > > A1: We kindly argue that our empirical analysis has included the so-called full learning. The gradient structure of DDNs per 10 epochs during training also support the conclusion for the full-learning procedure. Moreover, the loss is not that high. For example, the training loss is only ~1e-3 for the experiment of training LeNet on MNIST. This suggests that the pretrain models are indeed near minima.
> > >
> > > We observed the similar power-law gradient structure using two datasets in Figure 12, we believe that transfer learning is suitable for describing our main experimental settings. Because the gradient structure of pretrained models on a given dataset is different from the gradient structure in transfer learning. Our pretrained models are exactly the models which terminates at the final training phase. We pretrained the model and evaluated its gradient structure on the same dataset. If we are interested in the gradient structure during the initial phase of transfer learning, we should pretrain a model on one dataset (e.g. CIFAR-100) and evaluate the gradient structure on another dataset (e.g. CIFAR-10).
> > >
> > > R3: How are the first 2k out of 62k points significative of the tail of the distribution?
> > >
> > > A3: We respectful point out that the reviewer may unfortunately misunderstand the heavy-tail part of eigenvalues. In a heavy-tail distribution (https://en.wikipedia.org/wiki/Heavy-tailed_distribution), a heavy tail is a group of largest variables rather than small variables. Following classical related referees along this line of research, we choose study thousands of largest variables (/eigenvalues) larger than a bar (/a minimal cutoff value). This is not the exponential cutoff. Top thousands of largest eigenvalues are exactly the tails which we are interested in.
> > >
> > > Moreover, as the last paragraph of Appendix B.1 shows, ``First, focusing on relatively large values is very reasonable and common in various fields’ power-law studies (Stringer et al., 2019; Reuveni et al., 2008; Tang & Kaneko, 2020), as real-world distributions typically follow power laws only after/large than some cutoff values (Clauset et al., 2009) for ensuring the convergence of the probability distribution. Second, researchers are usually more interested in significantly large eigenvalues due to the low-rank matrix approximation.”
> > >
> > > R8. I find it confusing to state that the difference is in the first eigenvalues, when it is on the whole spectrum. I suggest rephrasing.
> > >
> > > A8: We apologized if we misunderstood the review’s point. We are afraid that we cannot directly say it may be generalized to the whole spectrum, as we only focused on and plotted the top thousands of eigenvalues, which is the tail part of the spectrum not the full spectrum. In contrast, previous works (Xie et al., 2020; 2022b) especially discussed the near-zero eigenvalues of Hessian and Covariance. Our observation on the tail part is not consistent with the previous observation on the near-zero part.

---

### Official Review · Reviewer_ZG6J · 2022-10-25

**Confidence:** 3
**Correctness:** 3
**Technical Novelty And Significance:** 2
**Empirical Novelty And Significance:** 2
**Recommendation:** 5

**Clarity, Quality, Novelty And Reproducibility:**

The results are clearly presented. The writing is good. The novelty seems to be marginal, the details can be found in the weaknesses part.

**Strength And Weaknesses:**

## Strength
- This work delivers thorough simulations to test the distribution of the stochastic gradients across both dimension and iteration. This is very important since, in the previous literature, there are two contradictory views about the distribution of stochastic gradients. The author first clarifies the difference between these two distinct views and put them into a consistent framework. Moreover, they do thorough simulations on many datasets, demonstrating the universality of their observations.

- The observation that gradient covariance has power-law tails seems very interesting, which might be able to explain the success of stochastic gradient methods.

## Weaknesses
- The main concern is that this work only provides some interesting observations, from which we cannot easily derive any informative implications. Hence, this work is more like a technical report instead of a paper. For example, they show that the gradient covariance has power-law tails in many neural network settings and they argue that this might be able to explain the success of stochastic gradient. It is not clear at all that why this special structure can help the generalization.

**Summary Of The Paper:**

This paper tests the distribution of stochastic gradient across both parameters and iterations. They show that dimension-wise gradients have power-law heavy tails. However, iteration-wise gradients and stochastic gradient noise caused by minibatch training have Gaussian tails. More interestingly, they show that the gradient covariance has power-law tails.

**Summary Of The Review:**

Please see the strength and weaknesses section.

---

> ### Author Response · Authors · 2022-11-18
> **Responses to Reviewer ZG6J**
>
> We sincerely thank Reviewer ZG6J’s comments and work.
>
> Q1: The main concern is that this work only provides some interesting observations, from which we cannot easily derive any informative implications. Hence, this work is more like a technical report instead of a paper. For example, they show that the gradient covariance has power-law tails in many neural network settings and they argue that this might be able to explain the success of stochastic gradient. It is not clear at all that why this special structure can help the generalization.
>
> A1: Our work makes a significant contribution to empirically and statistically rethinking the structure of stochastic gradients, which is a fundamental and essential issue in deep learning. Our work directly challenges some works on optimization and generalization that misused the structure/heavy tails of stochastic gradients. Thus, our contributions are novel and significant. We will discuss more relevant papers that are directly affected by our conclusion.

---

> > ### Comment · Reviewer_ZG6J · 2022-11-24
> > **Thanks for your response**
> >
> > Thanks for your response. I agree that your observations challenge many existing works on their gradient noise assumptions, but this has already been realized in the previous paper. What you have done is a systematic and rigorous test for the structure of the stochastic gradients. However, as an empirical paper, I still feel that if you want to claim that the special structure of stochastic gradient plays a critical role in the generalization of deep learning, more simulations are required to build such a relationship. Moreover, as a first step, having a good theory might not be necessary, but at least I would appreciate it if you could discuss what the intuition behind these observations is.

---

> > > ### Author Response · Authors · 2022-11-25
> > > **Responses to Updated Reviews of Reviewer ZG6J**
> > >
> > > We sincerely appreciate the reviewer’s constructive comments.
> > >
> > > Q2: I agree that your observations challenge many existing works on their gradient noise assumptions, but this has already been realized in the previous paper. What you have done is a systematic and rigorous test for the structure of the stochastic gradients.
> > >
> > > A2: We respectfully argue that our first contribution is indeed mentioned by previous papers but lacks evidences, which are essentially important for really reconciling the conflicted arguments. More importantly, our second contribution on the power-law covariance is very novel. Thus, our work present both systematic tests and novel findings.
> > >
> > > Q3: I still feel that if you want to claim that the special structure of stochastic gradient plays a critical role in the generalization of deep learning, more simulations are required to build such a relationship. Moreover, as a first step, having a good theory might not be necessary, but at least I would appreciate it if you could discuss what the intuition behind these observations is.
> > >
> > > A3: We argue that the special structure of stochastic gradients itself is a novel and significant contribution. A large body of papers (Li et al., 2020; Ghorbani et al., 2019; Zhao et al., 2019; Jacot et al., 2019; Yao et al., 2018; Dauphin et al., 2014; Byrd et al., 201 , Hardt et al., 2016; Wu et al., 2021; Smith et al., 2020; Wu et al., 2020; Sekhari et al., 2021; Amir et al., 2021) also support that the structure of stochastic gradients matters. We will make this point more clear in the revision.
> > >
> > > We admit that our work has not given a quantitative result based on the special structure. It can be very interesting to present intuitions or formal theoretical analysis in future. Fortunately, we have already obtained some interesting theoretical results and insights about the low-dimensional learning space of DNNs which has attracted much attention before. The power-law covariance may mathematically explain why training dynamics of DNNs happens in a tiny low-dimension space.
> > >
> > > We will add the relevant discussion and theoretical results into the revised paper, if the reviewer would like to support the acceptance. Here, we first present the draft in openreview. Please see: https://openreview.net/forum?id=9xlU4lhri9&noteId=gi87KDJ5wYm

---

### Official Review · Reviewer_2nic · 2022-10-29

**Confidence:** 3
**Correctness:** 3
**Technical Novelty And Significance:** 3
**Empirical Novelty And Significance:** 2
**Recommendation:** 5

**Clarity, Quality, Novelty And Reproducibility:**

Clarity:  The paper is written well and clear.
Quality:   The quality is good.
Originality:   Builds upon past work and reconciles/clarifies conclusions that are deviating in past literature may be due to the incoherent definition of SGN.
Reproducibility:  Adequate details are provided in the appendix. However, I am not absolutely sure about the speculations and quantitative claims in section 4 and 5.

**Strength And Weaknesses:**

Strengths:
The paper addresses an important problem. The experimentation with formal statistical tests and conclusions are interesting.

Weaknesses:
Apart from the fact that the authors conduct formal statistical tests (KS for power law and Chisquare test for Gaussianity) provide a little more rigor in the analysis, the central question of why these distributions arise is not addressed.  Indeed the authors point this out as a limitation and leave theoretical explorations to future work.  There are a number of points raised in the discussion section that are not completely resolved. For instance, why is the proportionality relation between the hessian and the covariance very weak?  The surprising discovery of invariance of the structure of the power law structure for SGN distribution is raised, but not analyzed.  I would have liked to see more of a concrete theoretical analysis of the problem with a simulated dataset involving a particular manifold structure and then studying the SGN structure thus allowing explicit correspondence between the nature of the dataset and the SGN distribution form.

**Summary Of The Paper:**

The effectiveness of Stochastic Gradient Descent optimization for deep learning is well known. The paper aims to study the nature of the probability distribution of stochastic gradient noise (SGN) via statistical tests that check for power law behavior or gaussian behavior of dimension-wise gradients, iteration-wise gradients, with mini-batch training.  The study uses MNIST, CIFAR10 datasets and LeNet, FCN, and ResNet18 architectures. The formal statistical tests reveal that dimension-wise gradients usually exhibit power-law heavy tails, while iteration-wise gradients and stochastic gradient noise caused by minibatch training usually do not exhibit power-law heavy tails. Power law distributions are reported for the covariance spectra of stochastic gradients.  The paper then speculates that this structure may be the reason for effectiveness of SGD in deep learning.

**Summary Of The Review:**

The paper is addressing a very important problem to unravel why SGD methods offer success in deep learning. Formal statistical tests for the distribution form of SGN and the empirical observation that the spectra of the covariance has a power law structure is interesting. However, the main difficulty I have with the paper is that I am unable to judge the empirical conclusions as there is limited explanation for the results and surprises. Moreover,  the authors themselves allude to the fact that the theoretical analysis is left for future work.

---

> ### Author Response · Authors · 2022-11-18
> **Responses to Reviewer 2nic**
>
> We gratefully thank Reviewer 2nic’s comments and hard work.
>
> Q1: The central question of why these distributions arise is not addressed.
>
> A1: It is true. We totally accept the reviewer’s opinion. It is very interesting and promising to address the theoretical origin of power-law gradients. However, we still argue that our work focuses on empirically and statistically rethinking the structure of stochastic gradients in deep learning. We discovered a number of interesting findings which cannot be explained by existing theories. The theoretical origins are beyond of the scope of our main contribution.
>
> Our contribution on empirical and statistical evidences is novel and significant. Because the structure of stochastic gradients is a fundamental and essential issue in deep learning, but many previous papers misunderstand it very much. Our work can help the papers which misunderstood or misused the heavy-tailed properties of stochastic gradients to fix the overlooked problems or reorganize novel results.

---

> > ### Comment · Reviewer_2nic · 2022-11-18
> > **Please comment about my other points**
> >
> > Thank you for your feedback. While I understand that theoretical justification may be left for future work, please comment about the soundness of your empirical conclusions as requested in my original review. Thanks.

---

> ### Author Response · Authors · 2022-11-18
> **Responses to “other points” of Reviewer 2nic**
>
> The reviewer mentioned ``there are a number of points raised in the discussion section that are not completely resolved’’.
>
> We kindly argue that our interesting empirical conclusions are clearly stated and totally based on the presented empirical and statistical evidences.
>
> We frankly admit that theoretical explanations are beyond the main contributions of this paper. The theoretical mechanism of this novel power-law structure is still unknown.
>
> About “other points”:
>
> Q2: For instance, why is the proportionality relation between the hessian and the covariance very weak?
>
> A2: Figure 5 directly presented the empirical evidence of the disabled proportionality between Hessian and Covariance (near minima). The very weak proportionality indicates the conventional theoretical approximation of two matrices, namely Eq. 3, can hardly hold along the sharp directions corresponding to large eigenvalues.
>
> Q3: The surprising discovery of invariance of the structure of the power law structure for SGN distribution is raised, but not analyzed.
>
> A3: Yes, our contribution is only the first to raise/report the surprising power-law structure of stochastic gradients. This finding is novel and significant. Again, we admit the theoretical origins are beyond of the scope of our main contribution.
>
> Q4: I would have liked to see more of a concrete theoretical analysis of the problem with a simulated dataset involving a particular manifold structure and then studying the SGN structure thus allowing explicit correspondence between the nature of the dataset and the SGN distribution form.
>
> A4: Again, we totally agree that it can be interesting or even very promising. Actually, we also want to see we can derive concrete theoretical results along this direction in near future.
>
>
> If the reviewer have other concerned points that we can address or discuss, we will be happy to address and discuss them.

---

> > ### Comment · Reviewer_2nic · 2022-11-21
> > **Thank you for your feedback!**
> >
> > Thank you for the clarifications.  I will factor this in my internal discussions with other reviewers.

---

### Author Response · Authors · 2022-11-25
**Theoretical Implication: Low-Dimensional and Robust Learning Subspace via Davis-Kahan Theorem**

We highly appreciate all reviewers for the constructive comments.

Fortunately, we have already obtained some interesting theoretical results and insights about the low-dimensional learning space of DNNs which has attracted much attention before. The power-law covariance may mathematically explain why training dynamics of DNNs happens in a tiny low-dimension space.

We will discuss the theoretical implications in the revised paper, if the reviewers support the acceptance. Here, we first present the draft in openreview.

---------------------------------------------------



[1] empirically observed that deep learning (via SGD) mainly happens in a low-dimensional space during the whole training process. [2] studied and reported that, throughout the optimization process, large isolated eigenvalues rapidly appear in the spectrum, along with a surprising concentration of the gradient in the corresponding eigenspace. [3] theoretically demonstrated that the learning space is a low-dimensional subspace spanned by the eigenvectors corresponding to large eigenvalues of the Hessian, because SGD diffusion mainly happens along these principal components. Note that the low-dimensional learning space implicitly reduces deep models' complexity. However, existing work cannot explain why the low-dimensional learning space is robust during training. In this paper, robust space means that the space's dimensions are stable during training.

We try to mathematically answer this question by studying the gradient covariance eigengaps. We define the $i$-th eigengap as $\delta_{k} = \lambda_{k} - \lambda_{k+1} $. According to Equation (2), we have $\delta_{k}$ approximately meeting

$\delta_{k} =  \Tr(C) Z_{d}^{-1} (k^{-\frac{1}{\beta-1}} - (k+1)^{-\frac{1}{\beta-1}}) = \lambda_{k} \left[ 1 - (\frac{k}{k+1})^{s} \right],$


where $Z_{d} = \sum_{k=1}^{n}  k^{-\frac{1}{\beta-1}}$ is the normalization factor. Interestingly, it demonstrates that eigengaps also approximately exhibit a power-law distribution when $k$ is relatively large. Particularly, we will have an approximate power-law decaying

$\delta_{k} =  \Tr(C) Z_{d}^{-1} (k+1)^{- (s + 1)} $


for eigengaps under the approximation $s \approx 1$. The power-law decaying eigengaps can be clearly verified by our additional experiments. Our experiments show that top eigengaps dominate others in deep learning similarly to eigenvalues.

Note that the existence of top large eigenvalues does not necessarily indicate their gaps are also statistically large. Previous papers revealed that top eigenvalues dominate others but did not reveal if top eigengaps dominate others in deep learning. Fortunately, we numerically demonstrate that, as rank order increases, both eigenvalues and eigengaps decay, following power-law distributions. Eigengaps even decay faster than eigenvalues due to the larger magnitude of the power exponent. We will show that this is the foundation of learning space robustness in deep learning.

\textbf{Eigengaps Bound Learning Space Robustness.} Based on the well-known Davis-Kahan $\sin (\Theta)$ Theorem \citep{davis1970rotation}, we use the angle of the original eigenvector $u_{k}$ and the perturbed eigenvector $\\tilde{u}_{k}$, namely $<u_{k} , \\tilde{u}_{k} >$, to measure the robustness of space's dimensions. We directly apply Theorem 1, a useful variant of Davis-Kahan Theorem [4], to the gradient covariance in deep learning, which states that the eigenspace (spanned by eigenvector) robustness can be well bounded by the corresponding eigengap.

------------------------
Theorem 1.Eigengaps Bound Eigenspace Robustness.

Suppose the true gradient covariance is $C$, the perturbed covariance is $\tilde{C} = C + \epsilon M$, the $i$-th eigenvector of $C$ is $u_{i}$ , and its corresponding perturbed eigenvector is $\tilde{u}_{i}$. Under the conditions of the Davis-Kahan Theorem and the power-law decaying eigengaps, the upper bound of $  \sin < u_{k} , \\tilde{u}_{k} >$ is given by

$2 \epsilon \|M \|_{op} (k+1)^{s+1}$,

where $ \|M \|_{op}$ is the operator norm of $M$.
------------------------

As we have a small number of large eigengaps corresponding to the large eigenvalues, the corresponding learning space robustness has a tight upper bound. Given the power-law eigengaps above, the upper bound of eigenvector robustness is relatively tight for top dimensions (small $k$) but becomes very loose for tailed dimensions (large $k$). This indicates that top eigenspace is stable for a small $k$ while non-top eignen space can be highly unstable during training a large $k$.

To the best of our knowledge, we are the first to demonstrate that the robustness of low-dimensional learning space directly depends on the gradient covariance eigengaps.

---

> ### Author Response · Authors · 2022-11-25
> **Reference**
>
> Reference:
>
> [1] Gur-Ari, G., Roberts, D. A., and Dyer, E. Gradient descent happens in a tiny subspace. arXiv preprint arXiv:1812.04754, 2018.
>
> [2] Ghorbani, B., Krishnan, S., and Xiao, Y. An investigation into neural net optimization via hessian eigenvalue density. In International Conference on Machine Learning, pp. 2232–2241. PMLR, 2019
>
> [3] Xie, Z., Sato, I., and Sugiyama, M. A diffusion theory for deep learning dynamics: Stochastic gradient descent exponentially favors flat minima. In International Conference on Learning Representations, 2021
>
> [4] Yu, Y., Wang, T., and Samworth, R. J. A useful variant of the davis–kahan theorem for statisticians. Biometrika, 102(2):315–323, 2015.

---

### Decision · Program_Chairs · 2023-01-20

**Decision:**

Reject

**Justification For Why Not Higher Score:**

As stated in meta-review

**Justification For Why Not Lower Score:**

N/A

**Metareview: Summary, Strengths And Weaknesses:**


The paper examines the structure of gradients in the training of deep neural networks. The authors focus on the heavy-tail hypothesis, which suggests that the heavy-tail properties of noise in the gradients contribute to the success of deep learning. Previous research on this topic has been divided, with some studies supporting the hypothesis and others refuting it.

The main contribution of the paper is the observation that whether or not the noise structure in gradients has heavy-tail properties depends on whether it is measured "across dimensions" or "across iterations" (see the paper for more details).

Reviewers generally agree that this contribution is interesting and novel. However, we have reached a consensus during the review process that the paper does not meet the standards for acceptance due to a lack of discussion about the implications of the observation and concerns about the significance of the paper. The authors have attempted to address these issues during the discussion period, but the issues are too significant to be resolved without another round of reviews.

Based on this, I must recommend the rejection of the paper at this stage. However, I hope that the comments provided will be helpful in improving the paper. Thank you for your submission.

**Summary Of Ac-Reviewer Meeting:**

N/A